# Social Innovation in Active Mobility Public Services in the Megacity of Sao Paulo

**Silvia Stuchi** [1,*] , **Sonia Paulino** [1] **and Faïz Gallouj** [2,*]

1 School of Arts, Sciences and Humanities, University of Sao Paulo, Sao Paulo 03828-000, Brazil
2 Faculty of Economics and Sociology, University of Lille, 59650 Lille, France
* Correspondence: silviastuchi@usp.br (S.S.); faiz.gallouj@univ-lille.fr (F.G.); Tel.: +55-11-94155-5993 (S.S.)

**Abstract:** This article aims to explore the relationship between social innovation and opportunities for innovation in public services, focusing on a range of initiatives intended to improve services and infrastructure for pedestrians in the city of São Paulo, the largest Brazilian megacity, namely: Reduced Speed Zone, Safe Routes to School, and Complete Street. We apply the multiagent framework for innovation in services, incorporating nine variables that characterize social innovation. As for the main results, in the local context, there is the role of third-sector organizations in creating and introducing solutions for active mobility services through co-creation. Co-creation was identified as a key process and is highlighted in actions to obtain community involvement, interviews to measure the acceptance of the project and detect potential points of improvement not foreseen in the pilot project, participatory workshops, installation of informative and interactive panels, preparation and approval of the temporary intervention project, and joint discussion and analysis with municipal agencies about the points that could receive the temporary intervention. The initiatives are recent and cover specific geographic–temporal boundaries. There is a need to deepen the dialogue between social innovation and service innovation with the co-design and co-construction approaches proposed in this paper, applied in different political, economic, urban, and social contexts. In addition, some barriers are highlighted relating to the lack of public funding, compliance with national regulations, political will, non-partisan actions, and long-term vision. There are potentials for the continuous introduction of innovations for the improvement of public services for pedestrians, promoting participatory restructuring as a form of (re)appropriation of urban public spaces.

**Keywords:** social innovation; co-creation; multiagent perspective

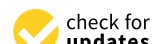



## 1. Introduction

Through active mobility initiatives located in the city of São Paulo, a Brazilian megacity, the article analyzes the relationship between social innovation and opportunities for innovation in public services. The selected experiences are: Reduced Speed Zone, Safe Routes to School, and Complete Street. Active transport can be defined as any form of human transport, such as walking, cycling, or wheelchairs, which uses the body to move, in some cases, relying on auxiliary devices [1]. Urban mobility is currently a hotly debated topic due to the COVID-19 pandemic, and several cities in the world have been betting on active mobility as a way to follow the recommendations of health agencies in daily commuting, through interventions that use tactical urbanism techniques, quickly expanding and improving infrastructure and services for active mobility. Tactical urbanism encompasses small-scale and short-term urban interventions projects, with the goal to inspire long-term transformation aiming to recover urban spaces. These interventions may take into account the participation of society [2,3], aiming to facilitate and encourage the testing of new concepts before being linked to important financial or political decisions, taking into account that the changes in question may or may not be accepted and/or adjusted, depending on the results obtained.

Walking, as an active mode of transport, comprises more than one third of trips in several cities around the world [1], and it can be better understood and analyzed from the "walkability" concept, encompassing infrastructure, access to goods and services, comfort and ambience, and attractiveness. Despite recent studies linking mobility and services, most notably the concept of 'MaaS' (Mobility as a Service) [4], our focus is on active mobility as a locus of service innovation aiming at meeting people's needs to move in public urban spaces, not only as a focus of technological innovation for the optimization of transportation networks as it occurs widely in the literature that looks at the logic of smart cities. Behind the focus on infrastructure, it is also necessary to consider public services. There is a technological and manufacturing bias in the conceptualization of service innovation, and in the traditional sense, it suffers from a number of shortcomings (mainly technology, manufacturing, and a market bias). For investigating the potential of innovation in services beyond the technologist view of innovation addressing sustainable urban mobility issues, we propose the use of a broader innovation approach, aiming to address sustainability, social inclusion, and social innovation [5–10].

Service innovation has a strong relationship with social innovation because service innovation can result from social innovation, i.e., social innovation can involve the provision of services to solve socio-economic issues [10], and both have similar characteristics such as multiactor collaboration and the development and implementation of new innovative solutions. Social innovations can bring a variety of contributions to society, such as improving living conditions in large cities through innovation in services focused on mobility and habitation [8,10,11]. Third-sector organizations, or social movements/ collectives, are often involved in social innovation since they ensure that society benefits from the outcomes of innovation [12,13]. These include regional and local urban transformations [14,15]. A variety of dimensions of sustainable urban transformations, across different scales, are identified: cities, buildings, neighborhoods, districts, villages, communities, and eco-urban projects [16].

Co-creation in social innovation focused on urban services is a recent area of study in the literature [7] and still needs to be investigated in more depth. Furthermore, given the intrinsic relationship between service and social innovation, the literature needs to be better linked and no longer treated as a separate subfield of study [10,17].

Public administration research, in particular in the context of the New Public Governance Paradigm [18–20], is considering the multiperspective/agent (or networked) approach to analyze outcomes and processes in public organizations, since, in line with the co-production literature [21], they involve different actors and levels of interaction. It should also be noted that social innovation and innovation in services are mostly studied in the European context, with a predominance of studies on innovation focused on technology and on the business sector. In order to collaborate towards filling this gap, we adapted, correlated, and made the necessary changes to the context of our geographic scope; we explore the relationship between social innovation and opportunities for innovation in public services for active mobility from a multiagent approach, with a focus on third-sector and public-sector participation in interventions in the pedestrian road area. Our scientific contribution is presented in the multidisciplinary and multiagent approach, which incorporates the innovation approach from the mobilization of service product characteristics (characteristics-based approach) in the search to broaden the discussion to also contemplate issues involving the public sector and the community, highlighting public services to requalify the urban public road space for active transport and promote sustainable urban mobility. Additionally, our research question is: how does the relationship between social innovation and opportunities for innovation in public service occur in active mobility initiatives?

After the Introduction, the paper is organized into three sections. Section 2 sets up the main concepts addressed and provides the theoretical framework of the paper. It discusses active mobility, social innovation, and the multiagent framework. Section 3 focuses on the empirical context and the methodology used. Section 4 displays the results, highlighting

the multiagent framework for service innovation incorporating the nine variables used to characterize social innovation (Figure 1).

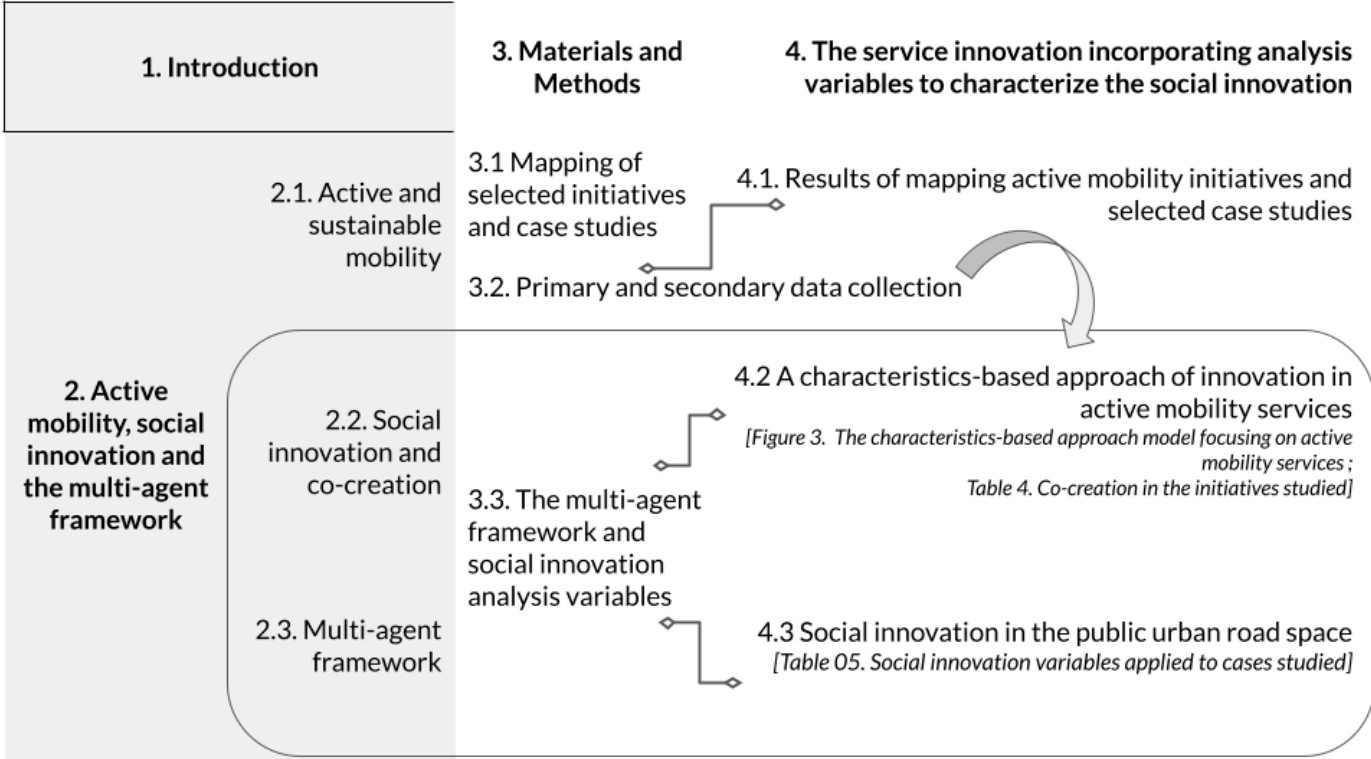

**Figure 1.** Systematic logic of the paper structure.

## 2. Active Mobility, Social Innovation, and the Multiagent Framework

This first section is devoted to a presentation and discussion of the main concepts that are at the basis of our analysis. We first address the notion of active mobility and its importance to achieve sustainability in cities. We then focus on social innovation in such a context. This innovation process involves a variety of actors which fall within the scope of a multiagent framework, the last point of our presentation.

### 2.1. Active and Sustainable Mobility

Active mobility (non-motorized transportation) includes walking, bicycling, wheelchair, and any means of transportation that uses human energy to move. Active transportation contributes to the environment and also generates positive impacts on the health and general well-being of the population [22,23]. Walking is the primary means of travel used by the world's population, as it is the most natural means of travel and connects all other means of transport—pedestrians are also users of public transport, thus every trip begins and ends on foot [22]. Yet, inversely to its importance, walking is often underestimated in the planning of urban mobility systems [24]. It is necessary to ensure safe, comfortable, and accessible routes for access to public transportation, especially in areas near stations and bus stops which have a high flow of pedestrians, which emphasizes the importance of public services to provide qualified access to users [24,25].

The consideration of active mobility is crucial for a sustainable urban mobility system, also promoting social inclusion and the right and access to the city. Considering that the transitions to sustainability are a significant opportunity to drive service innovation, one of the most prominent innovation domains, it is essential to have a comprehensive approach capable of incorporating the multiple and faceted socio-environmental and economic challenges and stakeholders [26,27].

Aligned with the Sustainable Development Goals (SDGs), established in the 2030 Agenda, it is worth noting that the climate issue is central to the action for cities that are more friendly and prepared for active mobility, aligned with the ambition of urban emissions targets that should be further intensified after the new IPCC report (2021). At the 21st meeting of the Conference of the Parties (COP21) to the United Nations Framework Convention on Climate Change, which took place in 2015, the goal of "sustainable transport" was introduced, but it was scarcely addressed at the recent COP 26 (2021). Active transportation combined with mass public transportation can help to mitigate some of humanity's most pressing environmental issues, such as climate emergency and air and noise pollution [1,24,25].

The role of innovation in urban issues is generally considered through the focus on "smart cities" [26]. Information and communication technologies are viewed as the foundation for contributing to improving the availability, quality, and accessibility of infrastructure and services, helping to bridge the gap between government and society (e-democracy and e-government/digital), with sustainable development serving as a backdrop. In order to broaden the understanding and conceptualization of the term "smart city", it is necessary to incorporate a variety of viewpoints (environmental, social, and economic). Additionally, given the complexities of these viewpoints and diverse contexts of cities around the world, there is no consensus on the principles that should be considered to build a smarter, more sustainable city [16,26,28]. Mobility is viewed as a tool for achieving a goal (through connecting people, commodities, services, and opportunities). As the aims of the same paradigm change, "smart" and "sustainable" are often tightly linked in this context [6,29]. We propose that the cases studied can be analyzed as opportunities for innovation in public services focused on pedestrian mobility linked to social innovation.

### 2.2. Social Innovation and Co-Creation

Social innovation incorporates different types of collective actions and social transformations that could potentially transform the "top down" economic society to a more participatory and "bottom up" society [21,30,31]. Social innovations can be defined by their social goals, which aim to improve the quality of life and well-being of communities and individuals, and by their (social) means [8,10,17,30]. The importance of user participation in the emergence and implementation of social innovation, as well as its local or popular nature and the relationship with sustainability, are all fundamental characteristics of social innovation [5,10]. The increased involvement of the third sector can be considered as social innovation per se [32,33].

Authors who address the role of citizens in public service provision have adopted some definitions of co-creation. By identifying user needs and behaviors, co-creation enables increments in service provision. We consider the definition of co-creation that presents degrees of citizen involvement in public services, categorizing them into intrinsic and extrinsic co-creation [21]. In intrinsic co-creation, there is co-construction, where citizens passively participate. In extrinsic co-creation, there are two types: co-creation and co-production. In co-creation, citizens actively participate through feedback and ideas. In co-production, citizens actively participate and are part of the implementation [21].

Co-creation can also bring positive results for the public sector, increasing transparency and efficiency, reducing costs and the risk of failure, enabling better adaptation to social needs, and consequently increasing user satisfaction [31,34]. We can mention other factors that impact the realization of innovation in multiagent contexts, as occurs in public–private partnerships. The main factors that influence the achievement of innovation in public–private partnerships are: structural factors, collaboration process factors, and participant-oriented factors [35].

The structural factor predetermines the structure and the environment of the collaboration process, contemplating the institutional spheres (political and regulatory) that impact the partners' actions and results. The collaborative process factor is a key aspect because public–private partnerships involve organizations with distinct interests, contemplating

the interaction between the actors involved in the initiatives. Finally, the participant-driven factor addresses the variables related to public and private actors' competences as well as the governance of the collaboration [35]. Some examples of these factors are presented in Table 1.

**Table 1.** Examples of factors that influence the achievement of innovation in public–private partnerships.

| Factors | Examples |
|---|---|
| Structural | - administration of contracts supporting performance and risk sharing<br>- assistance and participation of actors impacted by the innovation process |
| Collaborative process | - trust between the actors<br>- convergent objectives<br>- conflicting motivations and cultures |
| Participant-driven | - competences: technical know-how, professional background, knowledge in inter-organizational collaboration, and innovation training<br>- governance: public or private management can influence innovative results |

For this research, we highlight the relevant role played by third-sector organizations in the co-creation of social innovations at a local level. Underlining the multiagent and multidisciplinary approach in the initiatives studied, we can evidence the interaction of civil society with the public sector, aiming to improve walkability in the city.

*2.3. Multiagent Framework*

Environmental issues are complex, multifaceted, and multidisciplinary and cannot be solved consistently with a single type of ownership or institutional power.

There is the important assignment of participatory governance mechanisms in order to contemplate citizens in decision making that involves the distribution of public funds among communities, the formulation of public policies, and the monitoring and evaluation of government expenditures. In the National Policy for Urban Mobility (NPUM), non-motorized transport modes and collective public transport are prioritized, bringing up critical issues such as equity, sustainability and the involvement of society. In NPUM, there are collegiate committees, ombudsmen in organizations responsible for managing NPUM, mandatory public hearings and consultations, and systematic communication procedures in order to assess and monitor user satisfaction and strengthen public accountability [36].

A service can be described as the mobilization of internal or external competences, as well as internal or external techniques (tangible or intangible) to produce the product's final characteristics (goods or services) [37,38]—[Y] corresponds to the service characteristics; [T] to service provider technical characteristics; [T'] to clients/users technical characteristics; [C] to service provider competences; and [C'] to clients/users competences (Figure 2).

The model was revised in order to better integrate the agents, and in our cases studies highlighted the users, considered as associations of the third sector that represent interests in the active mobility field. The innovations that are based on the network configuration are a type of organizational innovation, with the understanding that the formation of the network can be interpreted as the target of innovation [37]. Consequently, the network configuration demands competences in the public sector that go beyond its organizational boundaries, incorporating practices and cultural characteristics of the collaborative partners, also demanding competences that involve aspects of communication, coordination, and governance.

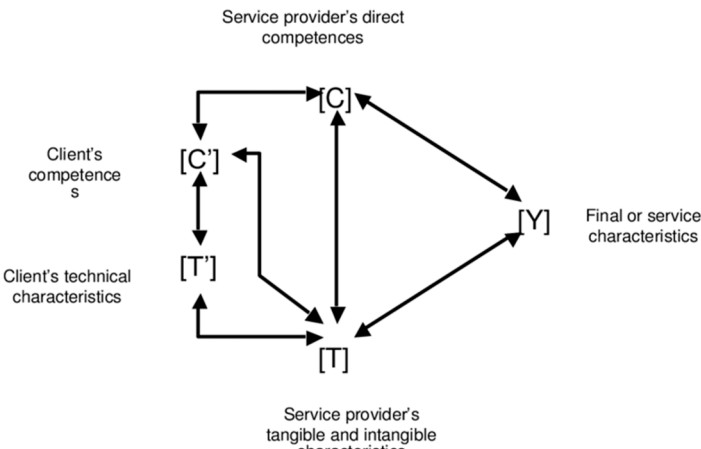

**Figure 2.** The product as a conjunction of vectors of characteristics and competences (characteristics-based approach).

## 3. Materials and Methods

Social innovation in urban areas is usually researched through case studies, using interviews, document analysis, and participant observation as a research method [31]. Our research methodology is also based on case studies, following three steps:

→ Mapping of selected initiatives and case studies.
→ Primary and secondary data collection.
→ The multiagent framework for service innovation incorporating analysis variables to characterize social innovation.

### 3.1. Mapping of Selected Initiatives and Case Studies

São Paulo, a megacity located in Brazil, presents challenges related to population commuting. Since 2014, the city has been developing tactical urbanism initiatives and has data, competence, and technical staff to adopt emergency measures and test the temporary expansion of the infrastructure for active mobility in these places aligned with the principles of the National Policy of Urban Mobility and the Municipal Urban Mobility Plan [39]. São Paulo is also one of the cities selected for the Bloomberg Philanthropies Global Road Safety Initiative, a program with the goal to reduce injuries and fatalities resulting from collisions worldwide. As part of the program, the Reduced Speed Zones, Complete Streets, and Safe School Route projects are being developed. The Bloomberg program was renewed for the period from 2020 to 2025.

Taking into account this context, we first performed a general mapping, addressed in the results section (Table 3) and, subsequently, the cases were selected based on the following criteria: developed in the city of São Paulo; in places with significant pedestrian mobility; connected to public transportation; and have used tactical urbanism techniques. The initiatives researched contribute to the improvement of active mobility, road safety, access to public transportation, and potentially contribute to the reduction in noise and air pollution.

### 3.2. Primary and Secondary Data Collection

For primary data collection, eight actors completed a semi-structured form. These actors are representatives of civil society, start-ups, universities that performed impact assessment diagnostics, and the public sector. In this amount of interviews, in the context studied, it is considered that all key players are contemplated, representing the categories of the model employed. It is not a matter of quantitative importance of interviewed actor, but of the role of relevance played by each of them in the case studies.

The objective of the survey was to investigate the changes in the services resulting from the implementation of the initiatives, bringing to light the description of the initiatives,

environmental improvements, participation of society and stakeholders that were contemplated in the initiatives, development of new methodologies and technologies focused on active mobility, technical training, and new competences and techniques acquired. For the secondary data collection, the following sources were used: diagnoses and reports on the initiatives, impact evaluation studies, Companhia de Engenharia de Tráfego (CET) database, particularly for the Reduced Speed Zones and the Life Protection Program [40–48]. The reports on interventions give information related to workshops, engagement processes with stakeholders, impact evaluations, intervention design, and materials applied.

*3.3. The Multiagent Framework and Social Innovation Analysis Variables*

In order to apply an innovation approach for analyzing the potential for innovation in services [37,49] that address sustainable urban mobility issues [26], including social innovation [5,10,30], we used: (1) the categorization we conduct, which is a result of the systematization and analysis of data from the vectors of service characteristics (Y), techniques (T), and competences (C) showed in topic 2.3 and, (2) based on the theoretical framework that characterizes social innovation, we propose 9 variables to analyze the active mobility initiatives, based on the recent literature addressing social innovation, service innovation, and multiagent configuration (Table 2).

**Table 2.** Variables used to analyze social innovation.

| | *Variables* | *References* |
|---|---|---|
| **1** | *Associations/collectives/activist groups:*<br>The relevant role of civil society | [5,10,14,27,30] |
| 2 | *Actions to highlight social needs:*<br>Methods, tactics, and tools used to gather information from the communities involved | [14] |
| 3 | *Third-sector interaction with the public sector:*<br>The means of third-sector interaction with the public sector/forms of third-sector participation to support the definitions related to the initiatives | [5,10,21,27,30,32] |
| 4 | *Evidence base:*<br>- Methodologies created by the public sector for the disclosure and validation of said techniques<br>- Participatory methodologies for disclosing and validating techniques | [16] |
| 5 | *Support from public policies for the diffusion of innovation:*<br>Public policies that are related to the contexts of the cases studied | [5] |
| 6 | *Government bodies that support the provision of services:*<br>Public bodies related to the projects, highlighting their interdisciplinarity and intersectionality | [27] |
| 7 | *Solution to respond to social needs:*<br>Local solutions, aimed at the population's well-being, resulting from the joint process of public-sector interaction and civil society participation | [5,10,14,27,30] |
| 8 | *Improvement of environmental quality:*<br>Aspects related to air and noise pollution | [10,14,27] |
| 9 | *Social impact:*<br>Developments achieved and measured by research methods | [10,14] |

Note: The italics correspond to the titles of the analysis variables.

The combination between the multiagent model, the characteristics-based approach, and the analysis variables aims to bring an approach that can integrate the dynamic interrelationships between stakeholders, the mechanisms of interaction and co-creation, and the techniques, competences, and impacts resulting from this system addressed in the results section, mainly throughout Figure 3. The characteristics-based approach model focuses on active mobility services. Table 4: co-creation in the initiatives studied and Table 5: social innovation variables applied to cases studied.

## 4. The Service Innovation Incorporating Analysis Variables to Characterize Social Innovation

Research in public services [12,13] has highlighted the collective preferences of citizens as beneficiaries of services and as participants in service creation, control, and planning. The

social innovation studies surveyed here can be analyzed in terms of a characteristics-based innovation approach focused on active mobility services in a multiagent context.

### 4.1. Results of Mapping Active Mobility Initiatives and Selected Case Studies

Active mobility initiatives were mapped (Table 3).

**Table 3.** Mapping of tactical urbanism initiatives in the municipality of Sao Paulo (Jardim Nakamura and Caminhar Pinheiros were not considered as case studies because of the unavailability of consistent data for analysis).

| Reduced Speed Zone | Implementation | Location | Tactical Urbanism |
|---|---|---|---|
| Bela Vista | April/2016 | Center | No |
| City center–1st and 2nd Phases | October/2013; December/2014 | Center | No |
| Consolaçao | June/2015 | Center | No |
| Lapa–1st and 2nd Phases | September/2014; March/2015 | West | No |
| Santana | September/2014 | North | Yes |
| Moema–1st and 2nd Phases | November/2014 | South | No |
| Penha | December/2014 | East | No |
| Bras | February/2015 | East | No |
| Sao Miguel Paulista | September/2015 | East | Yes |
| Pinheiros | November/2018 | West | Yes |
| Complete Street | Implementation | Location | Tactical urbanism |
| Joel Carlos Borges | September/2017 | South | Yes |
| Safe Routes to School | Implementation | Location | Tactical urbanism |
| Jose Bonifacio | May/2018 | East | Yes |
| Jardim Nakamura (There is no deep information available, unabling the case analysis) | June/2019 | South | Yes |

According to the criteria presented in Section 3.1, the studied initiatives are the following ones:

- Reduced Speed Zone (São Miguel Paulista and Santana): the main goal is to improve road safety for pedestrians and cyclists, while also improving environmental aspects. The maximum speed considered at the site is 40 km/h;
- Complete Street (Joel Carlos Borges): the main goal is the distribution of road space in order to accommodate the most vulnerable users, pedestrians and cyclists, with comfort and safety;
- Safe Routes to School (José Bonifácio): the main goal is to reduce the risk of traffic incidents, particularly in school zones.

### 4.2. A Characteristics-Based Approach to Innovation in Active Mobility Services

Although public services are often provided by public organizations, public services can be provided by public actors, private actors, or both. However, in all these different organizational forms, the public sector must ensure that this production occurs in an appropriate manner, assuming regulatory, inspection, incentive, and planning responsibilities. We suggest a representation of each agent's connection to the service vector, explained through the co-creation process (Figure 3). Co-creation requires more leadership from government [21], and the challenges are greater in developing and emerging economies, where information asymmetry, lack of accountability, transparency, and responsiveness to society (users' needs) are substantial.

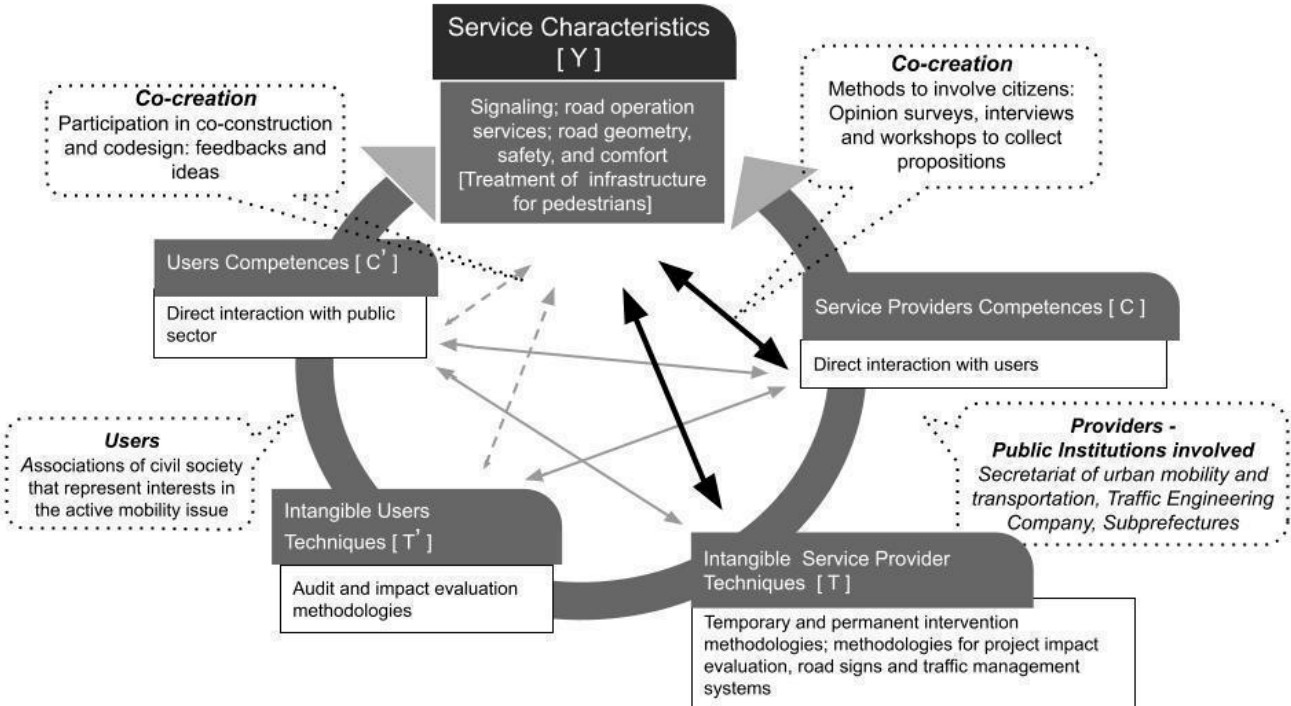

**Figure 3.** The characteristics-based approach model focusing on active mobility services.

The civil society associations related to active mobility issues represent the "users"; the public institutions that have acted in some way in the initiatives represent the "service providers" (the secretariat of urban mobility and transportation, the traffic engineering company-CET, and the subprefectures). In the light of the logic adopted in characteristics-based approaches to service innovation, social innovation does not consist of the technical artefact itself but of the additional service characteristics (functions) it embodies that have social and civic natures [10] (p. 6). Urban mobility encompasses not only infrastructure but also public services. The conditions of community access to transportation systems and infrastructure occur through the introduction and quality of public services that cover the road space. In [Y], we emphasize a multiagent approach, beyond infrastructure and technology, highlighting that while we take into account, for example, the construction of sidewalks and streets, similarly, we must consider the related public services (infrastructure treatment). New characteristics in public services were identified, i.e., improvement of signaling and road operation services, changing road geometry, and improving safety and comfort (infrastructure treatment), covering the following aspects: sidewalks, crossings, accessibility, horizontal and vertical signaling in public spaces, traffic light systems suitable for active mobility, and traffic moderation measures. These physical infrastructures support the additional service characteristics (functions) [Y]. In order to improve the quality level of the service offered to the population, the initiatives sought to increase safety and comfort for pedestrians by reducing speed limits, redesigning the road, and expanding sidewalks, road signs, and traffic management systems, i.e., micro-roundabouts; reduction in corner radii; sidewalk extension; and narrowing of traffic/moving lanes. The vector of technical characteristics (T and T′) is the set of techniques used to realize the product (good or service). Techniques are either tangible (computer hardware, machines, equipment, and other infrastructure items) or intangible (mathematical methods and work methods). Techniques are codified, transmissible, and independent of individuals. Tactical and temporary urban experiments are becoming more and more widespread as a technique [50] to face urban problems related, above all, to issues of public space and urban mobility—given the emergency that arises quickly for projects to be implemented in order to change the current urban situation.

The service providers intangible techniques (T) are summarized as the introduction of tactical urbanism practices: in accordance with the broad vision of social innovation, short-term and low-cost interventions attempt to stimulate grassroots restructuring, in a participatory approach, towards the re-appropriation of urban spaces by their own users, which extends potential forms of participation to specific actors, i.e., methods to involve citizens (opinion surveys, interviews, and workshops to collect propositions).

The users' intangible techniques (T') are summarized as the conduction of local audits and technical inspections to collect information and measure people flows, activities that contribute to the permanence of people in the space, road safety, aspects that involve the improvement of waste management, furniture, and project perceptions. This survey method collaborates with user participation, bringing information (qualitative and quantitative) about local community life. Impact evaluation methodologies (propensity score matching method) were identified, with the goal of evaluating the impacts of an intervention by comparing two groups: those who receive the intervention (factual) and those who do not (counterfactual). As the projects go beyond the "business as usual" approach of the Traffic Engineering Company (CET), methodologies for project impact evaluation were identified (volumetric counts, signaling, and road operation). Ex ante and ex post interviews were also conducted on temporary and permanent interventions in order to assess user perceptions of road geometry, safety, and comfort [Y], as well as interference between user displacements and economic impacts on the local market. Tangible techniques were not identified for service providers (T) and users (T').

The competences of the providers' (C) and users' (C') vector is the set of competences mobilized to perform the service, which can be scientific, technical, operational, and relational. They are mobilized to make the use of the techniques or the performance of the service accessible. Competences are embodied in individuals, a group of individuals, or an organization. Competences arise from training, experience, interactions, etc. They are tacit and difficult to transmit. In the case studied, we highlight the competences of service providers (back-office competencies) and competences that are necessary for the interaction of providers with users (user-facing competencies).

The service providers' competences (C) are identified through the engagement process carried out through the training workshops. Channels of direct interaction with users are opened, facilitating the exchange of information and tacit and non-tacit knowledge. Engagement and communication processes are central to the technique of tactical urbanism, as well as a key aspect of social innovation. This has led public authorities to acquire new competence of direct user interaction, which has strengthened the relationship of public authorities with society.

Users tend to act together through collaborative processes aiming to increase the societal capacity to act and to meet the sustainable social change goal [31]. Users' competences (C') are highlighted by working together with the public sector to design the temporary intervention project, to analyze, in collaboration with local agencies, which places might benefit from temporary intervention and to conduct community workshops (contemplating general users).

Recommendations were identified for new competences and techniques to be acquired for environmental impact assessment, both by the provider and the user. For the aforementioned evaluation, it was necessary to collect data on noise levels, vegetation cover, and air quality, all measurements should be performed ex ante and ex post interventions. Regarding environmental impacts, the São Miguel Paulista initiative tried to perform air quality measurement with a mini sensor called Bicycle Environmental Mapping-BeMap, which consists of measuring levels of CO and NOx and temperature and humidity data [43]. The data collection by BeMap was a pilot application and pioneer in Brazil. It has the advantage of being mobile and, therefore, has the potential to produce air pollution maps of the areas studied.

However, at the time, technical problems with the sensors and with the handling of the device prevented the collection of air pollution data and hindered the evaluation. As an

alternative, through a partnership with the laboratory of the Medical School of University of São Paulo, fixed stations were installed to collect particulate matter (MP2.5), which collected MP2.5 samples during three weeks in area 40 of São Miguel Paulista. However, because they are fixed stations, they are not able to capture variations in air pollution over the entire length of the analyzed area.

In relation to co-creation, in the cases studied, there was participation in co-construction and co-design through feedback and ideas (users) provided by methods to involve citizens (service providers). Co-creation cannot be considered as a purely "bottom-up" process, and organizing workshops is not sufficient to claim co-creation. It is necessary to have clear and well-structured methodologies to detect needs and co-design solutions [21]. In the analysis performed under these research boundaries, we can say that there is an effort of approximation with the co-creation processes but in lighter forms of co-creation, through co-construction and co-design. In the two last columns of Table 4, we present the iterative processes based on feedback from citizens often in collaboration with the public sector, and examples of new treatment services identified, based on co-creation processes involving the municipality, users, and third-sector organizations (Table 4).

**Table 4.** Co-creation in the initiatives studied.

| Initiative | Co-Creation (Similarities) | Iterative Processes: Based on Feedback from Citizens, Often in Collaboration with the Public Sector [21] | New Characteristics in Public Services (Improvement of Signalling and Road Operation Services, Changing Road Geometry, and Improving Safety, Comfort, and Traffic Moderation Measures) |
|---|---|---|---|
| José Bonifácio | Co-design/co-construction | Participatory meetings, idea workshops, and opinion polls were conducted with the surrounding communities to collect proposals for the intervention's site | Micro-roundabout offset curb extensions/chicane; sidewalk extensions; narrow traffic/moving lane; new crosswalk |
| São Miguel Paulista | Co-construction | The participation of citizens occurs in a passive way, without effective awareness of their role; information about the intervention was collected from local residents through interviews, but no participatory workshops were held. | Sidewalk extensions; mini-squares; narrowing of traffic/moving lane; and a crosswalk |
| Santana | Co-design/co-construction | Training for the municipal technicians; idea workshops with the population; and presentation at the Trade Association on the day of the intervention, comprising educational and cultural activities, panels, and banners for feedback and ideas | Micro-roundabout; reduction in corner radii; sidewalk extensions; and narrowing of traffic/moving lane |

In the context of the initiatives studied, despite the limitations for conducting a more profound analysis, the importance of co-creation is emphasized in the ability to involve citizens, contemplating their experiences and capabilities, because citizens have a unique perspective on the quality of public services—since they are service users. Thus, opinion surveys and also participatory idea workshops were conducted with the communities to collect proposals for changes at the site of intervention.

Although the idea workshops tried to contemplate different audiences, it was observed that the meetings that proposed to involve the surrounding community, in relation to the totality of the local population, did not obtain a satisfactory amount of public attention. On one hand, there is the hypothesis of reduced interest and/or lack of community involvement, and on the other hand, there is the process of communication and information about the events that could be improved to attract more participants, for example, trying to provide more opportunities for different schedules. Even with these constraints, the comments from the participants were favorable to the proposed interventions.

Workshops were held in order to identify challenges, potentials, and opportunities in the area of the initiatives and also aimed to insert the population in the decision-making process. The meetings sought to take into account the diversity of the participants and ensure that everyone was included in the discussion. These interactive meetings presented a greater diversity of agents when a forum was provided, with the conduction by civil society supported by the public sector. Experts and technicians also participated in order to cooperate with the urban requalification project, and the involvement of users (residents and neighborhood visitors) occurred in order to expand the knowledge about concepts

related to the urban project, such as road safety, traffic calming, etc. The population consultations that took place after the interventions were less effective in terms of the general public's acceptance of the modifications.

On Joel Carlos Borges Street, despite the data collection carried out with the local community, the project continued without clear and coordinated communication with the population. In November 2017, questionnaires were administered to street merchants and users in an effort to better understand the impacts of the project. Additionally, the data collection aimed to investigate the transportation and consumption behaviors of people accessing the street.

In the José Bonifacio initiative, questionnaires were administered in public schools in 2018 by the public sector, supported by ITDP Brazil, the Bloomberg Initiative, and NACTO. However, there is a participation gap for identifying proposals for intervention with the local community of Jose Bonifacio (i.e., participatory workshop). Even with the involvement of civil society organizations, it should be noted that in Jose Bonifacio, the public sector played a more prominent role in terms of interaction and the use of tactical urbanism as a test phase of the urban project.

*4.3. Social Innovation in the Public Urban Road Space*

For the researched active mobility initiatives, the following results are presented: distribution of the use of public circulation space, roads, and public areas; prioritization of non-motorized transport modes over motorized ones and of public collective transport services over individual motorized transport; and mitigation of environmental, social, and economic costs of displacement of people in the city. The studied initiatives, developed through cooperation agreements between city hall and civil society organizations, are presented in Table 5 in light of the nine analysis variables that characterize social innovation.

There is also the role of organized civil society [7,9,10,31,39] carrying out urban audits and impact evaluations, evaluating the performance of public bodies, working together with the city hall and the temporary intervention project, analyzing with the public sector the places that could receive the temporary intervention, and holding workshops with the community. Tactical urbanism initiatives were carried out through cooperation agreements between the public sector and civil society, pointing to the structural factors contributing to risk management and sharing [35].

The role of organized civil society is reduced in the Safe Route to School intervention, in which, through the experiences acquired from previous interventions, the public sector, represented by CET, develops and leads the process of engagement of the local community and the intervention of tactical urbanism. There is great potential for improving public services for non-motorized transport in line with the paradigm of sustainable urban mobility based on social innovation.

The public service for active mobility is not yet explicitly considered in official documents, with data and information that allow analysis. A major focus on infrastructure is observed, while it is necessary to address and understand services. In some of the objectives formulated in these texts (e.g., "improvement of integration conditions with the different modes"), it seems that the service dimension is implicit. The prevailing misconception is that providing infrastructure is sufficient to ensure that mobility services in the city are adequately provided. This will not happen if it is not accompanied by adequate public services (treatment of infrastructure for pedestrians).

**Table 5.** Social innovation variables applied to cases studied.

| Cases | Santana Reduced Speed Zone | São Miguel Paulista Reduced Speed Zone | Joel Carlos Borges Complete Street | José BonifácioSafe Route to School |
|---|---|---|---|---|
| Service characteristics: Signaling and road operation services (Treatment of infrastructure for pedestrians) | Micro-roundabout; reduction in corner radii; sidewalk extensions; narrowing of traffic/moving lane | Sidewalk extensions; mini-squares; narrowing of traffic/moving lane; crosswalk | Sidewalk extensions; narrow traffic/moving lane | Micro-roundabout offset curb extensions/chicane; sidewalk extensions; narrow traffic/moving lane; new crosswalk |
| Timeframe (year) | Implementation: 2014 Temporary and permanent interventions: 2017 and 2018 | Implementation: 2015 Temporary intervention: 2016 Permanent intervention: has not occurred | Implementation, temporary, and permanent interventions: 2017 | Implementation, temporary, and permanent interventions: 2018 |
| 1. Associations/collectives/activist groups | Bloomberg Initiative (BIGRS), Global Cities Design Initiative (NACTO-GDCI), and WRI Brasil and Vital Strategies (and in the impact evaluation study: FGV/CEPESP and ITDP) | Bloomberg Initiative; Transport and Development Policy Institute (ITDP), NACTO and WRI Brasil, Vital Strategies and iRap/GRSF | WRI (and in the impact evaluation studies: Cidade Ativa, Urb-i, WRI, Labmob UFRJ, and Metrópole 1: 1) | Bloomberg Philanthropies, NACTO, Vital Strategies, and ITDP |
| 2. Actions to highlight social needs | Activities to stimulate community participation; interviews to evaluate the perception and acceptance of proposals and identify potential for improvement in the pilot project of the interventions; workshops with the active involvement of the participants; implementation of informative and interactive panels | | | |
| 3. Third-sector interaction with the public sector | Design and approval of the temporary intervention project; analysis with the public sector about possible locations for temporary intervention; meetings with the community; and cooperation agreements with the municipality | | | There was joint discussion and analysis, however, in this initiative, a relevant role of the public sector was identified based on the experiences acquired with previous interventions. |
| 4. Evidence base | Carrying out tests with the mini-roundabout to be implemented; diagnosis of user perception and evaluation; and urban reading methodology | An evaluation approach that goes beyond pedestrian and vehicle volumetric counting, performed in partnership with civil society, aiming for a systemic view of the uses, local dynamics, and behavioral aspects of users | | From the experiences acquired with previous interventions, development of a methodology that systematizes actions focused on improving the safety for pedestrians by CET |
| 5. Support from public policies for the diffusion of innovation | Life Protection Program, Safe Pedestrian Program, implementation of areas with reduced maximum speed, Municipal Plan for Urban Mobility, Pedestrian Statute, Sidewalk Emergency Plan (PEC), Cyclef Plan, Master Municipal Plan Law, and Zoning Municipal Law | | | |
| 6. Government bodies that support the provision of services | Mobility and Transport Secretariat, Traffic Engineering Company (CET), and sub-prefectures | | | |
| 7. Solution to respond to social needs | New urban design with requalification and redistribution of road space; speed reductions were enforced; and priority for active mobility | | | |
| 8. Improvement of environmental quality | Especially in the aspects related to air quality and noise pollution reduction, the initiative reports indicate a tendency of the interventions to improve environmental quality. However, until now, no consistent information and evaluations are available to ratify these improvements. | | | |
| 9. Social Impact | Definition of temporary stages to measure impacts and results; the organizations Cidade Ativa and LabMob UFRJ were responsible for carrying out impact evaluations (local diagnosis on the flow of people and vehicles, opinion polls, and analysis of users' habits and profiles) | | | Definition of temporary stages to measure impacts and results; the Traffic Engineering Company was responsible for conducting the impact evaluation |

## 5. Conclusions

In light of social innovation, we highlight the promotion of new perspectives for the transformation of urban public road spaces aligned to decision making in favor of a sustainable and equitable urban mobility system, consistent with the National Policy of Urban Mobility. Participatory governance involves designing new forms of collaboration and partnerships between public, social, economic, knowledge, and civic actors. It is important to note that the conditions of negotiation, responsibilities, empowerment, and connections between actors from different sectors vary widely. It is emphasized that attention to arbitration by public-sector actors is part of the process of change.

In the local context, the role of volunteers and collectives is highlighted in the design and introduction of innovative solutions for active mobility, responding to the demands of society and giving priority to pedestrians in the use of urban road spaces. These organizations focus on the well-being and needs of individual citizens and use their experience to act and interface with public service providers, identifying areas of intervention and establishing evidence bases for the success of the strategies used to requalify road spaces.

Social innovation is centered on the interaction between agents and embraces the co-creation of new services/products. In this research, driven by civil society organizations, we identified co-construction and co-design supported mainly by opinion surveys and participatory idea workshops. Co-creation is highlighted in actions to obtain community involvement, interviews to measure the acceptance of the project and detect potential points of improvement not foreseen in the pilot project, participatory workshops, installation of informative and interactive panels, preparation and approval of the temporary intervention project, and joint discussion and analysis with municipal agencies about the points that could receive temporary intervention.

The study provides elements for the discussion of public policies in line with the challenges of sustainable urban mobility in the 21st century, amplified by the COVID-19 pandemic, and attuned to social and environmental needs and adequate public services to prioritize active modes of transport.

As the initiatives are recent and cover specific geographic–temporal boundaries, for future research aiming to deepen the dialogue between social innovation and service innovation, with the co-design and co-construction approaches proposed in this paper, similar approaches need to be applied in different political, economic, urban, and social contexts, (e.g., Europe, North America, Asia, and other South American countries). The impacts of innovation seen through case studies focused on urban services could be better understood through a continuous monitoring and evaluation process and with the establishment of comparative bases.

**Author Contributions:** Conceptualization, S.S., S.P. and F.G.; methodology, S.S. and S.P.; validation, S.S., S.P. and F.G.; formal analysis, S.S.; investigation, S.S.; resources, S.S.; data curation, S.S. and S.P.; writing—original draft preparation, S.S., S.P. and F.G.; writing—review and editing, S.S., S.P. and F.G.; visualization, S.S., S.P. and F.G.; supervision, S.P. and F.G.; project administration, S.S. and S.P.; funding acquisition, S.S. and S.P. All authors have read and agreed to the published version of the manuscript.

**Funding:** This research is supported by Coordenação de Aperfeiçoamento de Pessoal de Nível Superior-Brasil (CAPES)-Finance Code 001.

**Institutional Review Board Statement:** Research project number 2018/877 was approved by the Research Committee of the School Arts, Sciences and Humanities/University of São Paulo/.

**Informed Consent Statement:** Not applicable.

**Data Availability Statement:** Not applicable.

**Conflicts of Interest:** The authors declare no conflict of interest.

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
