# Peer review of "Social Innovation in Active Mobility Public Services in the Megacity of Sao Paulo"

_sustainability, doi:10.3390/su141911834_

Round 1

Reviewer 1 Report

The authors propose explore the relationship between social innovation and opportunities for innovation in public services, focusing on a range of initiatives intended to improve services and infrastructure for pedestrians in the city of São Paulo.

The authors present the motivation and problems in section introducción, but lack a paragraph that indicates "our scientific contribution is xxxx".

A methodology is presented in a clear and orderly manner, which consists of 3 steps:

i) Mapping of selected initiatives and case studies

ii) Primary and secondary data collection

iii) The multi-agent framework for service innovation incorporating analysis  variables to characterize the social innovation

I will recommend to modify the section 3, the explanation of the steps of the activities must be separated from the results

I will recommend to modify the paper as follows: 1) Abstract should be written as 1) objective, methodology, 3) findings, 4) conclusion, and 5) implications.

Author Response

Response to Reviewer 1 Comments

Point 1:. The authors propose explore the relationship between social innovation and opportunities for innovation in public services, focusing on a range of initiatives intended to improve services and infrastructure for pedestrians in the city of São Paulo. The authors present the motivation and problems in section introducción, but lack a paragraph that indicates "our scientific contribution is xxxx".

Response 1:  We inserted a paragraph that indicates our scientific contribution: “our scientific contribution is presented in the multidisciplinary and multi-agent approach, in the search to broaden the discussion to also contemplate issues involving the public sector and the community, highlighting public services to requalify the urban public road space for active transport and promote sustainable urban mobility”

Point 2:  A methodology is presented in a clear and orderly manner, which consists of 3 steps:

  1. i) Mapping of selected initiatives and case studies
  2. ii) Primary and secondary data collection

iii) The multi-agent framework for service innovation incorporating analysis  variables to characterize the social innovation

I will recommend to modify the section 3, the explanation of the steps of the activities must be separated from the results

Response 2:  we understand that the suggestion is applicable only to topic 3.1 and we have moved the mapping results to the results section. The topics 3.2 and 3.3 were already dedicated to content related to methodology, and the results of the application of these instruments are discussed mainly in topics 4.2 and 4.3 ( previously topics 4.1 and 4.2)

Point 3: I will recommend to modify the paper as follows: 1) Abstract should be written as 1) objective, methodology, 3) findings, 4) conclusion, and 5) implications.

Response 3: The abstract was reformulated: 

“This article aims to explore the relationship between social innovation and opportunities for innovation in public services, focusing on a range of initiatives intended to improve services and infrastructure for pedestrians in the city of São Paulo, the largest Brazilian megacity: Reduced Speed Zone, Safe Routes to School, and Complete Street. We apply the multi-agent framework for innovation in services, incorporating 09 variables that characterize social innovation. As main results, in the local context, there is the role of third sector organizations in creating and introducing solutions for active mobility services, through co-creation.  Co-creation was identified as a key-process and is highlighted in actions to obtain community involvement, interviews to measure the acceptance of the project and detect potential points of improvement not foreseen in the pilot project, participatory workshops, installation of informative and interactive panels, preparation and approval of the temporary intervention project, and joint discussion and analysis with municipal agencies about the points that could receive the temporary intervention. The initiatives are recent and cover specific geographic-temporal boundaries. There is a need to deepen the dialogue between social innovation and service innovation, with the co-design and co-construction approaches proposed in this paper, applied in different political, economic, urban, and social contexts.  In addition, some barriers are highlighted: related to the lack of public funding, compliance with the National regulations, political will, non-partisan actions and long-term vision.  There are potentials for the continuous introduction of innovations for the improvement of public services for pedestrians, promoting participatory restructuring as a form of (re)appropriation of urban public spaces”

Reviewer 2 Report

The research question is important: "the relationship between social innovation and opportunities 9 for innovation in public service"l

I recommend to add a flowchart showing the logical links between different sections of the paper because I lost the main point many times.

English needs to be improved: using shorter and simpler sentences would help.

More articulation and justifications on the used methodology seem necessary. 

Author Response

Response to Reviewer 2 Comments

Point 1:. The research question is important: "the relationship between social innovation and opportunities  for innovation in public service"

Response 1:  The research question is: how does the relationship between social innovation and opportunities  for innovation in public service occur in active mobility initiatives? 

Point 2: I recommend to add a flowchart showing the logical links between different sections of the paper because I lost the main point many times

Response 2:  A flowchart was added in order to show the logical links between different sections of the paper (Figure 1. Systematic logic of the paper structure)

Point 3:. English needs to be improved: using shorter and simpler sentences would help More articulation and justifications on the used methodology seem necessary. 

Response 3: we hope that all the modifications performed, taking into account also the comments of the other 2 reviewers, covered this remark as well. 

Reviewer 3 Report

This article discusses the topic of social innovation in the field of active mobility with reference to interventions to improve public mobility services. The study is set up methodologically through the analysis of case studies framed in a multi-agent framework that considers the mutual interactions between different stakeholders, their respective competencies, and characteristics.

The article is well structured, the objectives are clearly defined. The methodology adopted, although clearly pointed out, needs more detailed framing, just as some of the key elements that underlie the work, sustainable mobility and active mobility, need to be framed more solidly in the vast literature on the topic.

The following are some detailed observations.

-       The first lines of the introduction (lines 26-29) are identical to those of the abstract: the abstract should not be a fragment of the introduction, and vice versa.

-       The concepts of active mobility and sustainable mobility are introduced and defined in a short paragraph (2.1). Since these are key elements for the article and, in general, topics that have been covered extensively and articulately in the literature, a more robust and extensive bibliographic framing would be appropriate.

-       Figure 1 should be illustrated more clearly, and in particular all the symbols and concepts in it should be defined (in fact, many of them are defined only in Figure 2 below). Also, it is not clear why all the relationships between the different elements that appear in the diagram are bidirectional.

-       Section 3.1 specifies that the selected cases are characterized by the use of tactical urbanism techniques. It is necessary to explain, including with literature references or examples, what is meant by "techniques" of tactical urbanism.

-       In line 257, the vectors Y, T, and C (found in Figure 1) are mentioned and explicit reference is made to "topic 1.3": what is it?

-       A sentence like that in 291-294 needs to be substantiated by appropriate literature references. There are many mobility studies, a vast literature, that are not infrastructure-based at all (think, for example, of analyses of user mobility behavior, acceptance of mobility technologies, quality of public transportation services, etc...). It is clear what the authors intend to say in this passage, but it needs to be argued in a more robust way. Infrastructure and technology are always present in mobility studies, the difference lies in the fact that in some cases they are in the background, in other cases they have a prominent position and are the subject of the analyses and research.

-       The first lines of the conclusions (429-436) seem more like a continuation of the comments in the previous paragraph and should be re-arranged.

-       The conclusions contain general considerations, which are also correct and appropriate, while the article refers to specific case studies in Brazil. It would be interesting to highlight, even on a general level, how differences in regulations and approaches to the implementation of sustainable mobility urban plans in different parts of the world (e.g., Europe, North America, Asia, South America) can be reconciled with the co-design and co-construction approaches proposed in this paper.

Author Response

Response to Reviewer 3 Comments

This article discusses the topic of social innovation in the field of active mobility with reference to interventions to improve public mobility services. The study is set up methodologically through the analysis of case studies framed in a multi-agent framework that considers the mutual interactions between different stakeholders, their respective competencies, and characteristics. The article is well structured, the objectives are clearly defined. The methodology adopted, although clearly pointed out, needs more detailed framing, just as some of the key elements that underlie the work, sustainable mobility and active mobility, need to be framed more solidly in the vast literature on the topic. The following are some detailed observations.

Point 1:.  The first lines of the introduction (lines 26-29) are identical to those of the abstract: the abstract should not be a fragment of the introduction, and vice versa.

Response 1:  This paragraph was reviewed. 

Point 2:     The concepts of active mobility and sustainable mobility are introduced and defined in a short paragraph (2.1). Since these are key elements for the article and, in general, topics that have been covered extensively and articulately in the literature, a more robust and extensive bibliographic framing would be appropriate.

Response 2:  We discussed and presented more references about these key elements, as follows:

[22] De las Heras-Rosas, C. J.; Herrera, J. Towards Sustainable Mobility through a Change in Values. Evidence in 12 European Countries, Sustainability, vol. 11(16), pages 1-23, 2019. https://doi.org/10.3390/su11164274

[23] Lee, S. Satisfaction with the Pedestrian Environment and Its Relationship to Neighborhood Satisfaction in Seoul, South Korea. Sustainability 2022, 14, 9343. https://doi.org/10.3390/su14159343

[24] Appolloni, L.; Corazza, M.V.; D’Alessandro, D. The Pleasure of Walking: An Innovative Methodology to Assess Appropriate Walkable Performance in Urban Areas to Support Transport Planning. Sustainability 2019, 11, 3467. https://doi.org/10.3390/su11123467

[25] Lopes Toledo, A.L.; Lèbre La Rovere, E. Urban Mobility and Greenhouse Gas Emissions: Status, Public Policies, and Scenarios in a Developing Economy City, Natal, Brazil. Sustainability 2018, 10, 3995. https://doi.org/10.3390/su10113995.

Point 3:  Figure 1 should be illustrated more clearly, and in particular all the symbols and concepts in it should be defined (in fact, many of them are defined only in Figure 2 below). Also, it is not clear why all the relationships between the different elements that appear in the diagram are bidirectional.

Response 3:  We explained Figure 1 in more detail “([Y] corresponds to the service characteristics, [T] to service provider technical characteristics; [T'] to clients/ users technical characteristics; [C] to service provider competences; and [C'] to clients/ users competences)”, in addition, we developed a deeper explanation after figure 3 (named as figure 2 before revision), the adaptation of figure 1 for our study. 

Point 4:. Section 3.1 specifies that the selected cases are characterized by the use of tactical urbanism techniques. It is necessary to explain, including with literature references or examples, what is meant by "techniques" of tactical urbanism.

Response 4:  We developed a deeper explanation, including the reference Cariello et al. (2021), combining it with our theoretical approach of services and justifying the use of tactical urbanism as a technique. As follows: 

“The vector of technical characteristics (T and T′) is the set of techniques used to realize the product (good or service). Techniques are either tangible (computer hardware, machines, equipment, and other infrastructure items) or intangible (mathematical methods, work methods). Techniques are codified, transmissible, and independent of individuals. Tactical and temporary urban experiments are becoming more and more widespread as a technique [50] to face urban problems related, above all, to issues of public space and urban mobility - given the emergency that arises quickly for projects to be implemented in order to change the current urban situation.”

Additional reference: [50] Cariello, A.; Ferorelli, R.; Rotondo, F. Tactical Urbanism in Italy: From Grassroots to Institutional Tool—Assessing Value of Public Space Experiments. Sustainability 2021, 13, 11482. https://doi.org/10.3390/su132011482

Point 5:.  In line 257, the vectors Y, T, and C (found in Figure 1) are mentioned and explicit reference is made to "topic 1.3": what is it?

Response 5:  it was corrected. The correct relation is “topic 2.3”

Point 6:. A sentence like that in 291-294 needs to be substantiated by appropriate literature references. There are many mobility studies, a vast literature, that are not infrastructure-based at all (think, for example, of analyses of user mobility behavior, acceptance of mobility technologies, quality of public transportation services, etc...). It is clear what the authors intend to say in this passage, but it needs to be argued in a more robust way. Infrastructure and technology are always present in mobility studies, the difference lies in the fact that in some cases they are in the background, in other cases they have a prominent position and are the subject of the analyses and research.

Response 6:  We agree with your remarks and explanation. In order to better explain this idea, we rewrite the paragraph: “Urban mobility encompasses not only infrastructure, but also public services. The conditions of community access to transportation systems and infrastructure occur through the introduction and quality of public services that cover the road space. In [Y], we emphasize a multi-agent approach, beyond infrastructure and technology, highlighting that while we take into account, for example, the construction of sidewalks and streets, similarly, we must consider the related public services (infrastructure treatment).”

Point 7:. The first lines of the conclusions (429-436) seem more like a continuation of the comments in the previous paragraph and should be re-arranged.

Response 7:  It was re-arranged.

Point 8:.  The conclusions contain general considerations, which are also correct and appropriate, while the article refers to specific case studies in Brazil. It would be interesting to highlight, even on a general level, how differences in regulations and approaches to the implementation of sustainable mobility urban plans in different parts of the world (e.g., Europe, North America, Asia, South America) can be reconciled with the co-design and co-construction approaches proposed in this paper.

Response 8:  we consider it a very important consideration to be explored in depth in future work. Thus, we have enhanced the paragraph which indicates future research opportunities: “As the initiatives are recent and cover specific geographic-temporal boundaries, for future research aiming to deepen the dialogue between social innovation and service innovation,with the co-design and co-construction approaches proposed in this paper, similar approaches need to be applied in different political, economic, urban and social contexts, (e.g., Europe, North America, Asia, and other South American countries)”

Round 2

Reviewer 2 Report

This comment was ignored by the authors:

More articulation and justifications on the used methodology seem necessary.

Author Response

Response to Reviewer 2 Comments

Dear Reviewer, 

Thanks for identifying the weaknesses in our paper and providing us with the opportunity to strengthen our manuscript. 

Please find below our answers:

Point 1:. This comment was ignored by the authors: More articulation and justifications on the used methodology seem necessary.

Response 1:    In order to address this comment, we highlight the following aspects that were added and developed in the first round of review:  

A research question was included and a flowchart was added in order to show the logical links between different sections of the paper (Figure 1. Systematic logic of the paper structure)

In order to solve this gap saliented by the reviewer, in this second round of revisions, in addition, aligned also with the articulation and justifications about the methodology used, we sought to expand the clarifications in the following excerpts: 

.

- a paragraph that indicates our scientific contribution, also presenting the articulation and justifications on the used methodology: our scientific contribution is presented in the multidisciplinary and multi-agent approach,that incorporates the innovation approach from the mobilization of service product characteristics (characteristics-based approach), in the search to broaden the discussion to also contemplate issues involving the public sector and the community, highlighting public services to requalify the urban public road space for active transport and promote sustainable urban mobility

- We explained Figure 1 in more detail “([Y] corresponds to the service characteristics, [T] to service provider technical characteristics; [T'] to clients/ users technical characteristics; [C] to service provider competences; and [C'] to clients/ users competences)”, “In [Y],  we emphasize a multi-agent approach, beyond infrastructure and technology, highlighting that while we take into account, for example, the construction of sidewalks and streets, similarly, we must consider the related public services (infrastructure treatment). In addition, we developed a deeper explanation after figure 3

- The combination between the multi-agent model, the characteristics-based approach, and the analysis variables aims to bring an approach that can integrate the dynamic interrelationships between stakeholders, the mechanisms of interaction and co-creation, the techniques, competences and impacts resulting from this system, addressed in the result section, mainly throughout  Figure 3.  The characteristics-based approach model focusing on active mobility services ; Table 4. Co-creation in the initiatives studied; and Table 05. Social innovation variables applied to cases studied. 

- We add in table 4 the connections concerning the methodology used, highlighting examples of new treatment services identified, based on co-creation processes involving the municipality, users and third sector organizations:

We hope that these clarifications were sufficient to answer this important comment made by the reviewer. 

Should you have any questions, please let us know. 

King regards

Reviewer 3 Report

The article has been significantly improved, and the suggestions addressed to the authors during the first round of review have been taken up. In particular, the methodological frame and state of the art have been strengthened and are now presented more clearly.
The article in my opinion is ready for publication, I suggest only a final check for English verification and minor typos.

Author Response

Dear Reviewer, 

Thanks for identifying the weaknesses in our paper and providing us with the opportunity to strengthen our manuscript.

We provided a final check for English verification and minor typos.

Should you have any questions, please let us know. 

King regards, 

Round 3

Reviewer 2 Report

Well done!